# Linking Superior Developmental Feedback with Employee Job Satisfaction? A Conservation of Resources Perspective

**DOI:** 10.3390/ijerph20043211

**Published:** 2023-02-12

**Authors:** Zhongqiu Li, Haoqi Qin, Xue Zhang, Qiwen Zhang, Linshan Tang

**Affiliations:** 1College of Economics and Management, Northeast Agricultural University, Harbin 150030, China; 2College of Philosophy, Law and Political Science, Shanghai Normal University, Shanghai 200233, China

**Keywords:** superior developmental feedback, employee resilience, job satisfaction, job complexity

## Abstract

Previous studies have shown that superior developmental feedback (SDF) has a mixed impact on employees’ long-term development, but its effect on job satisfaction (JS) has been generally ignored. Therefore, this study proposes and tests a model based on the conservation of resources theory to shed light on how feedback from a leader or superior may increase employees’ JS. In this study, researchers analyzed responses from a two-stage questionnaire distributed to 296 employees to test the proposed hypotheses using MPlus 7.4 software. The results show that employee resilience (ER) partially mediates the link between SDF and JS. The results also indicate that the relationship between SDF and ER is strengthened by job complexity (JC). The results provide novel avenues for further study and practice in the areas of SDF and JS.

## 1. Introduction

The association between superiors’ behavior and employees’ attitude toward their work has been demonstrated in numerous studies over the past 10 years [1,2,3,4,5] (Gilboa, Shirom, Fried, and Cooper, 2008; Jackson and Schuler, 1985; O’Driscoll and Beehr, 1994; Tubre and Collins, 2000). Recently, researchers have increasingly extended the research into superiors‘ behavior by examining the employee’s psychology with measures such as JS as the outcome [6,7]. As a crucial component of an employee’s psychology, JS refers to employees’ perceptions of job pleasure as well as their emotional assessments of those perceptions [8,9]. Communication between superiors and employees is the key to influencing employee JS [2]. As a basic communication strategy between employees and superiors [10], feedback can profoundly impact employees’ perceptions and behaviors [11,12]. SDF, as a typical superior–employee communication, is defined as positive behavior in the workplace through which superiors provide helpful or valuable information to support employees’ learning, development, and job improvement [13]. Digging deeper into the mechanism of SDF on employee JS can not only enrich and expand the research on the positive influence of superiors’ feedback behaviors on employees; it also helps to solve the contemporary proposition of how superiors exert influence on easing employees’ psychology in significant public health emergencies.

The primary objective of our research was to explain the association between SDF and JS through the conservation of resources (COR) theory. Unlike other types of feedback, SDF is designed to convey information that can help employees to progress, and it aims to enable them to grow and develop in their work [13]. SDF does not set mandatory goals for employees and SDF allow employees to have more freedom in establishing their own goals [13]. Employees generally appreciate comments from their superiors, because this can help them to improve both their performance and the efficiency of their organization [14]. Therefore, employees would rather connect with and receive recognition from those in higher-ranking positions inside the organization than with and from their colleagues [15,16]. However, recent studies in organizational behavior have confirmed the reality of SDF and investigated its beneficial effects on employees’ behaviors [17,18]. A more in-depth and methodical investigation into how and when SDF influences employees’ JS needs to be conducted. In recent studies, the effect of SDF on employees has been analyzed from an interactionist point of view [19], social exchange theory [17], and social cognitive theory [18], but these studies have ignored the important role of feedback as a resource. As suggested by COR theory [20], resources in the workplace can emanate from the empathetic actions of leaders such as through SDF, which employees perceive as a route to achieving their objectives. Thus, SDF fits well with the basic assumptions of COR theory. We also focus on the mediating role of JC between SDF and JS. JC enhances employees’ ability to cope with their jobs under adverse conditions and to recover from turbulent and stressful jobs [21]. Thus, this research adopts a novel viewpoint, the COR theory, which both strengthens the connection between SDF and JS and broadens the theory’s potential applications.

In addition, to obtain a deeper understanding of the SDF–JS relationship, our research tries to discover a key boundary condition of the causal sequence. The use of ER, as a construct that embodies positive adaptive characteristics when one is facing adversity, can contribute to the development of research that explores the mechanisms of JS formation in the context of COVID-19 by examining its mediating role in the relationship between SDF and the JS of employees. Recent research suggests that job characteristics and their effects must be considered when superiors influence their employees [22]. COR theory indicates that in challenging situations, individuals need additional resources from their environment [23]. For this reason, this study views high JC as an unprecedented challenge to an individual. Then, we test the moderating effect of JC on relationship between SDF and ER. Thus, ER will be stronger for employees with high JC. This study generally investigates the mechanism of SDF that affects employee JS and its boundary conditions according to COR theory. This can provide a theoretical basis and practical guidance for enterprise managers on using leaders’ feedback to restore employees and thus enhance their positive work attitudes.

In the following sections, we first develop the relationship between SDF and employee JS based on COR theory. Second, we investigate the relationship between ER and employee JS and further develop a mediation assumption that links SDF and employee JS via ER. Third, we argue that the proposed consequence is stronger in environments with high levels of JC. Fourth, we test the research model using the data of 296 Chinese employees. We conclude by describing the theoretical and practical contributions of our research to the SDF-, JC-, and JS-related literature.

## 2. Theoretical Background and Hypothesis Development

Resources play an essential role in combatting the adverse effects of loss or potential loss. COR theory suggests that individuals will strive to retain, protect, and construct resources to avoid resource loss [23]. On the one hand, individuals invest their own resources into mitigating stressful events and preventing the experience of adverse consequences. On the other hand, individuals invest resources to protect themselves from potential losses. Supportive behaviors in the workplace can be considered valuable resources by employees [21,24]. Resources for employees may come from useful information provided by superiors on their work-related behaviors, such as through SDF. COVID-19 has added numerous uncertainties to people’s personal and work lives, which can easily pose a potential threat to a company and its employees. Currently, it is particularly critical for individuals in the work environment to obtain resources through others. By seeking help from others, individuals are motivated to obtain additional resources to compensate for the imbalance between their capabilities, needs, and psychology [25]. Therefore, this study constructs a mechanism for the effect of SDF on employee JS and concludes that SDF affects employee JS through the mediating role of ER. It is also hypothesized that the mediating role of ER varies somewhat among individuals with different levels of JC.

### 2.1. SDF and JS

Drawing from COR theory and the characteristics of SDF, this study suggests that SDF can enhance employee JS. SDF, as positive leadership behavior, can provide employees with signals of increased resources, which in turn can enhance their JS. The signals of resource changes received by employees in the workplace can significantly impact their attitudes and behaviors [20]. As the most important environmental factor in the employee’s job, the supervisor is an essential source of energy for the employee. COR theory suggests that resources are interrelated, and that resources aggregate to protect individuals from excessive demands or stressful events as well [26]. Positive psychologists argue that supportive leadership is an important working resource for enhancing behavioral motivation [27]. SDF can save employees’ energy by providing them with useful, evaluative, and forward-looking information while allowing them to maintain their personal career development aspirations [13].

Employees perceive SDF as a precious resource, since their leaders are continuously imparting useful information through their communications, emails, and interactions with them in the workplace. It has been shown that employees feel more energized when they interact positively with their superiors [28]. When individuals receive supportive psychological resources from the context in which they are located, their JS is significantly enhanced [29]. COR theory suggests that good interactions with others can help employees to obtain additional resources [30]. Additionally, SDF focuses on the future development of employees and can be viewed as a positive resource by employees. When superiors treat employees benevolently and caringly, the employees tend to show positive attitudes toward their organization [31,32]. It has been shown that employee JS emphasizes the individual’s cognitive evaluation of the quality of job wellbeing and is affected by factors such as superiors, colleagues, and salaries [33,34]. When leaders have frequent contact with their employees in instances in which employees are provided with positive feedback, this benign communication gives employees more support and resources, which will lead to more promotion opportunities and rewards, a reduction in threats from the organization, a weakened perception of job dissatisfaction, and an enhanced JS [35]. In conclusion, four hypotheses are proposed in this study, the first of which is:

**Hypothesis** **1.**
*SDF is positively related to employee JS.*


### 2.2. Mediating Effects of ER

ER, as a personal resource, can enable employees to handle challenging circumstances, difficulties, failures, and potential threats [21,36,37]. ER is often viewed as an individual trait, an attribute, a process, a kind of psychological capital, and a skill [21]. It has been shown that individuals can acquire personal resources through training and constant development, which includes ER [38]. This study proposes that ER has a mediating role in the relationship between SDF and employee JS. The main reasons for this are as follows. First, the supportive behaviors of superiors facilitate ER [39]. Further, COR theory assumes that an individual’s resources do not exist in isolation and can be acquired through cultivation, learning, and adaptation; that is, resources are generated by the interaction between the individual and the environment or organization [40]. When employees believe that their superiors can provide them with useful information—that is, developmental feedback—the relationship between employees and superiors is stronger, and employees will have more access to critical resources in the organization, such as information that is valuable for their job or development [13]. This means not only that employees will perceive an increased likelihood of their own professional success, but also that they will have some confidence in being able to quickly compensate for lost resources in the event of a potential threat or failure.

Second, SDF can be seen as an environmental condition that contributes to the creation of individual employee resources. COR theory suggests that the resources available to individuals and groups tend to coalesce with each other, creating a “caravan channel” that serves as an environmental condition to maintain, enhance, and protect individual resources [41]. Therefore, when superiors provide support for technical and emotional challenges through encouraging behaviors, employees are more likely to recover their resources quickly, that is, via ER [21]. Second, the essence of ER as a type of psychological capital lies in adaptation, especially when facing adversity. According to COR theory, individuals with high ER can maintain adequate resources and focus on tasks in the face of significant challenges, as well as approach challenges at work in a positive and complete way [26,42], thus allowing employees to better adapt to their jobs, complete their work tasks, and increase their JS. Employees’ JS reflects an individual’s evaluation of multiple aspects of their job. Individuals with high ER are proactive in preparing for difficulties, minimizing the impact of current or potential stressors on their work [43], and attenuating the negative impact of potential threats. Therefore, individuals with high ER are more likely to evaluate aspects of their job as satisfactory. In conclusion, the research proposes the following hypothesis:

**Hypothesis** **2.***ER mediates the relationship between SDF and employee JS*.

### 2.3. Moderating Role of JC

The job characteristics of employees are significant factors influencing the role of superiors [22]. JC, as a typical job trait, refers to the intricacy of the specific content of the job [44], which is reflected in the quantity, quality, and difficulty of the tasks in terms of job requirements. JC is multifaceted and requires employees to combine knowledge from multiple sources and imposes additional challenges on employees [45]. JC can be viewed as a form of adversity or challenge for employees, and the variable, uncertain, complex, and vague context under a public health emergency exacerbates the uncertainty of job tasks and increases the degree of JC. COR theory indicates that when confronting major challenges, individuals need more resources to be able to cope, and when individual resources are gradually consumed, more resources need to be obtained from the environment [23,46]. SDF provides employees with general information, focusing on the employee’s improvement [13,19]. The positive effect of SDF on ER is amplified in individuals with high JC. When JC is high, it is difficult for employees to complete required tasks on their own, thus enhancing the importance of feedback from others. In contrast, the positive effect of SDF on ER is weakened when employees find that the level of JC is low, the job does not require excessive knowledge and skills, and employees do not need to obtain help and resources from others. Previous studies have found that preferential treatment and support given to employees by superiors makes them feel cared for, and that employees can obtain material assistance and emotional support in times of need [47,48]. Employees who are asked to accomplish more than they are capable of may seek out additional resources to make up for their potential losses, such as in their level of energy and self-esteem. In situations of high JC, employees who perceive that their abilities are not sufficient to meet challenges at work will attempt to obtain additional resources from their environment. It has been shown that leaders in organizations play an important role in the resilience of their employees when the latter face adversity [21]. This is a situation in which valuable information from superiors (i.e., SDF) regarding the job and the employee’s development becomes particularly important. Collectively, JC positively moderates the effect of SDF on ER, and ER further transmits this effect to JS. Thus, JC positively moderates the indirect effect of SDF on employee JS by enhancing ER.

Therefore, the following hypotheses are proposed:

**Hypothesis** **3.**
*JC moderates the relationship between SDF and ER, such that there will be a stronger positive relationship when JC is high.*


**Hypothesis** **4.**
*The positive indirect effect of SDF on employee JS—via ER—is moderated by JC, so that the indirect effect is stronger when JC is high.*


This study constructs a mechanism of the influence of SDF on employee JS and concludes that SDF affects employee JS through the mediating role of ER. It is also hypothesized that the mediating role of ER varies somewhat among individuals with different degrees of JC. A theoretical model of these ideas is shown in Figure 1.

## 3. Materials and Methods

### 3.1. Participants and Procedure

With the assistance of human resource managers, we randomly distributed questionnaires to 400 full-time employees from technology companies in northeast China online. Data were collected at the beginning of June 2021, and the return of the questionnaires was completed in July 2021. Samples were collected from different geographical regions of China, including the cities of Harbin, Beijing, Shenyang, Chongqing, and Fushun, for richer sampling. A critical incident methodology was used to test participants’ reactions to specific events in a general survey [49].

We collected data in two waves to greatly reduce the common method bias (CMB) [50]. At Time 1, the distributed questionnaires had three parts. The first part was a short introduction to the content of the questionnaire and the uses of the questionnaire data. The second part was mainly for control variables, intended for the preliminary screening of respondents and the collection of their basic information, such as their gender, age, education, and tenure. The third part was the main part of the questionnaire, which contained the measurement scales for three variables: SDF, ER, and JC. At Time 2, two weeks after the completion of the first questionnaire study, employees were asked to rate their JS.

The total number of questionnaires distributed at Time 1 was 400, and 361 questionnaires were collected. At Time 2, questionnaires were distributed to subjects who completed the questionnaires at Time 1, and 307 questionnaires were recovered. The researchers matched the two questionnaires and finally synthesized the complete questionnaires. After excluding the questionnaires that were obviously not carefully completed, in which there were missing data, or questionnaires that could not be matched, 296 valid questionnaires remained, and the valid questionnaire recovery rate was 74.00%. The basic information of the sample is shown in Table 1. The proportion of male employees and female employees in the valid sample was the same; in terms of education, those with bachelor’s degrees accounted for the highest proportion of 59.12% of the valid sample; those aged 26–35 were in the majority, accounting for 63.44%; and those with less than five years of service were in the majority, accounting for 56.08% of the valid sample.

### 3.2. Measurement

All scales used in this study were derived from previous research published in reputable journals, guaranteeing their accuracy. Standard translation and back-translation processes were used to translate the measurement items utilized in English into Chinese [51]. Two bilingual doctoral students were chosen to translate the English questionnaire into Chinese, and later, one lecturer and two doctoral students were selected to translate the Chinese questionnaire back into English. After completing these steps, 10 employees from different companies were asked to participate in a pre-survey to ensure the accuracy of the questionnaire’s language. The wording of the questionnaire was appropriately modified to avoid distortion to ensure that it matched the expression habits of employees. A classic 5-point Likert scale was used (1 = strongly disagree; 5 = strongly agree) in the survey questionnaire, and all materials were presented in Chinese.

SDF: We used the three-item scale developed by Zhou [13] to identify SDF. Sample items included: “While giving me feedback, my supervisor focuses on helping me to learn and improve”, and “My supervisor provides me with useful information on how to improve my job performance”. The Cronbach’s alpha was 0.84.

ER: ER was measured using the six-item scale developed by Smith et al. [52]. A sample item read: “I usually come through difficult times with little trouble”. The Cronbach’s alpha was 0.83.

JC: JC was measured by the three-item scale adapted from Shaw and Gupta [53]. Sample items included: “My job requires a high level of skill”, and “My job requires a lot of mental effort”. The Cronbach’s alpha was 0.72.

JS: This variable was measured using the scale developed by Liu et al. [54], which includes 3 items, some of which were: “In conclusion, I am satisfied with my job”, and “In general, I like working here”. In the present study, the Cronbach α coefficient of this scale was 0.88.

Control variables. Previous studies have shown that age, organizational tenure, gender (0 = female, 1 = male), and education (1 = have a junior college degree or below, 2 = have a bachelor’s degree, 3 = have a master’s degree, and 4 = have a doctoral degree) could potentially influence employee perceptions and behaviors [55,56,57]. Therefore, the above variables were controlled for in this study.

## 4. Results

### 4.1. Analytical Strategy and Preliminary Analysis

The study used SPSS 26.0 and Mplus 7.4 for statistical analysis. First, validation factor analysis was used to test the construct validity, and descriptive statistics and stratified regression analysis were performed to test the hypotheses with the help of SPSS 26.0 software. To ensure the credibility of the results, the mediating effects and moderated mediating effects were tested with the help of the PROCESS plug-in using bootstrapping.

The results of the validation factor analysis are shown in Table 2. The fit indices of the four-factor model were better than those of the other factor models (χ^2^ = 102.98, df = 48, χ^2^/df = 2.15, CFI = 0.97, TLI = 0.96, RMSEA = 0.06, SRMR = 0.05). It is evident that there is good discriminant validity among the variables in this study.

Since SDF, ER, JC, and JS are self-rated by employees, CMB may exist. In this study, the severity of the CMB deviation was tested using Harman’s one-way method. The results indicated that the first factor explained only 29.56% of the variance, and the problem of CMB in this study was acceptable.

### 4.2. Descriptive Statistics and Correlation Analysis

The mean, standard deviation, and correlation coefficient of each variable and the results of descriptive statistical analysis are shown in Table 3.

For this study, the commonly used Pearson correlation coefficient was used to reflect the correlation between the variables. The results in Table 3 reveal that SDF is positively associated with JS (r = 0.38, *p* < 0.01), and ER positively affects JS (r = 0.58, *p* < 0.01).

### 4.3. Hypothesis Testing

Hierarchical regression analysis was used to test the hypothesis in this study, and the results are presented in Table 4. In the first regression equation (Model 4), the control variable was the independent variable, and JS was used as the dependent variable. In the second regression equation (Model 5), the control variable and SDF were jointly used as independent variables, and the dependent variable was JS. There was a positive effect of SDF on JS (β = 0.37, *p* < 0.001), so Hypothesis 1 was verified.

Hypothesis 2 (see Table 4) states that ER mediates the relationship between SDF and JS, which was also supported. SDF positively affects JS (β = 0.37, *p* < 0.001). Model 6 shows that ER positively affects JS (β = 0.64, *p* < 0.001). The relationship between the independent variable (SDF) and the dependent variable (JS) was still significant when the mediating variable (ER) was added to the regression model, but the effect became weaker (from β = 0.37, *p* < 0.001, to β = 0.19, *p* < 0.001). To further investigate whether ER can play a mediating role between SDF and JS, with the help of bootstrapping repeated sampling, a random sample was repeated 5,000 times, and the numerical results were calculated. The results showed that ER had a partial mediation effect on the link between SDF and JS with the help of SPSS PROCESS (indirect effect = 0.172, S.E. = 0.037, 95% C.I. = 0.106 to 0.248). Therefore, Hypothesis 2 was verified.

For Hypothesis 3, the results (see Table 4) revealed that the JC had a significantly positive moderating effect on the relationship between SDF and ER (β = 0.008, *p* < 0.01). To gain insight into the moderating role of JC, it was plotted in Figure 2.

As seen in Figure 2 above, the overall level of ER is higher when individuals have high levels of JC than it is for individuals with low levels of JC. In contrast, the relationship between SDF and ER becomes weaker when JC is low. Thus, Hypothesis 3 was validated.

For Hypothesis 4, we calculated the conditional mediating effect of ER at different levels of JC. Specifically, the indirect effect of SDF on JS through ER was stronger when JC was higher (b = 0.232, boot S.E. = 0.048, 95% bias-corrected CI = (0.148, 0.330)) than when JC was lower (b = 0.144, boot S.E. =0.047, 95% bias-corrected CI = (0.148, 0.330)). The index of moderated mediation was significant (b = 0.067, S.E. = 0.034, 95% C.I. = 0.002 to 0.134). Therefore, Hypothesis 4—that J.C. has a positive moderating effect on the indirect effect of SDF on JS via ER (i.e., the higher the degree of JC, the stronger the indirect effect)—was verified.

## 5. Discussion

It is the environment that is uncertain, and the enterprise itself that is certain. Facing environmental uncertainty, companies need to be prepared to enhance ER, strengthen employees’ psychological constitution, and improve employee JS, thus promoting the sustainable development of companies under major public health emergencies. SDF, a common strategy used by superiors to manage employees, can have a significant impact on employee perceptions and behaviors. To reveal the mechanism by which SDF affects JS under major public health emergencies, this study constructed a first-stage mediated effects model based on COR theory, using ER as a mediating variable and JC as a moderating variable. The empirical analysis of the two-stage data of 296 employees led to the following conclusions.

First, this research contributes to the feedback literature by offering an additional account to discern the relationship between SDF on JS from the perspective of resource conservation. Current research on supervisors’ influence on employees’ behavior has focused mainly on supervisors’ traits [58], and the theoretical basis of research on SDF is more limited. Although previous studies have related SDF to employees’ cognition and behavior from an interactionist point of view [19], social exchange theory [17], and social cognitive theory [18], the linkage from SDF to JS, as well as the mediating effect of ER, have not been established. COR theory expands the scope of interpretation for the effect of SDF, which regards SDF as a resource for employees, thus providing a new perspective to explain the relationship between SDF and employee JS and enriching and expanding the findings on the influence of SDF on employees’ psychology. Our findings agree with the findings of positive psychologists who have proposed that supportive leadership is an important job resource for increasing employee JS [59]. The attention given to employees, as reflected in the SDF and the valuable information provided by it, can be considered an important resource by employees. SDF fits well with the basic assumptions of COR theory, and this study helps to reveal the positive effects of SDF by explaining the effects of SDF on employees’ work attitudes according to COR theory.

Second, ER mediates the relationship between SDF and JS. This study’s findings have, to some extent, opened the “black box” of the effect of SDF on JS. Although it has been shown that supervisors’ feedback can positively predict employee JS [60], the internal mechanism of this relationship is not clear in the existing research. ER can help individuals to recover from adversity or negative emotional experiences and effectively “bounce back” to their original state [21], thereby enhancing their ability to cope with adverse conditions. To cope with adversity in the environment (e.g., JC), the social environment and the superiors in the organization play an important role regarding ER [61]. Drawing from COR theory, this study points out the important role of ER as a bridge between SDF and JS. This assertion not only reveals how SDF affects employee JS from a resource source conservation perspective, but it also adds to the studies of Srivastava and Madan [60] on the influence of leadership feedback on JS.

Third, JC positively moderates the role of ER in mediating the relationship between SDF and JS. This finding clarifies the boundary conditions of the mechanism of the role of SDF in influencing employee JS. This study proposes the moderating role of JC from the perspective of job characteristics, responding to the call of Han et al. [22] to consider differences in job characteristics and their effects on the process of superiors’ exertion of influence. This study empirically concludes that even when facing adversity (JC), employees can build a “resource caravan” by obtaining resources from others (i.e., through SDF) [61].

This study also has some practical implications. First, organizations should encourage supervisors to concentrate on employees’ development because of the beneficial effects of supervisor development feedback on employees. Nowadays, the competition among enterprises is full of variables and uncertainties, and job satisfaction of employees, as the overall attitude of employees towards their work content and work environment, is also increasingly a concern of superior decision makers. To promote employees’ growth, superiors should give employees more options for training and development guidance on how to advance in their careers, as well as advice on how to do a better job. In management practice, enterprises should create a good working environment for employees, provide sufficient work resources, including developmental feedback, and then increase employees’ psychological resources. Second, considering the important role of ER, individuals with higher ER show better JS than individuals with lower ER. Therefore, employees with higher ER are better prepared for difficult times. Individual resilience at work is not immutable, and ER can be cultivated [38]. Companies need to strengthen the psychological development of their employees when facing the changing environment of the times [62]. Therefore, organizations and superiors should pay attention to the development of ER. For example, they could organize ER training programs to enhance employees’ individual competencies and strategies to cope with challenges, changes, and ambiguities, which in turn helps to improve their JS. Finally, superiors should focus on the individual attributes of their employees when interacting with them and differentiate their management style according to individual differences to better utilize the positive effects of SDF.

## 6. Conclusions

Increasing attention has been paid to research on the antecedents of employees’ JS. The present study investigates the relationship between SDF and employees’ JS by shedding light on the mediating role of ER and the moderating role of JC. Our research helps to establish a stronger evidence-based understanding of the positive influence of SDF. We expect to forge a stronger and more precise linkage between SDF, individual differences, and employees’ psychology and to provide both organizational scholars and human resource practitioners with meaningful insights on ways to use feedback and manage JS in organizations effectively.

Our study also has several limitations. First, all the data used in the study were obtained from the Chinese context, which plays a limiting role in the external validity of the study’s findings. Correspondingly, cultural dimensions (e.g., collectivism) should be acknowledged to understand how employees feel and react to SDF with JS. For example, when collectivism is high, subordinates may be more likely to have a positive attitude toward their organization because of supportive feedback in the workplace. Consequently, further research could reexamine the findings of this study in other cultural contexts to test its cross-cultural generalizability. Second, although the selected sample involved different regions, the sample size was limited; therefore, future research needs to be improved by further expansion of the sample size. Third, the study only collected samples from high-tech industries, and the question as to whether the findings of this study can be used in other industries is yet to be investigated. Third, although we collected data from multiple regions and set a time-lag design, there may still be reverse causality in the current data. For example, individuals with high JS may have high levels of SDF. In future studies, it will be necessary to conduct longitudinal and experimental designs to enhance the causal relationships of the model.

## Figures and Tables

**Figure 1 ijerph-20-03211-f001:**
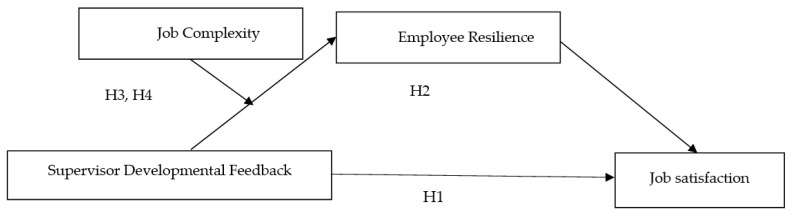
Research model.

**Figure 2 ijerph-20-03211-f002:**
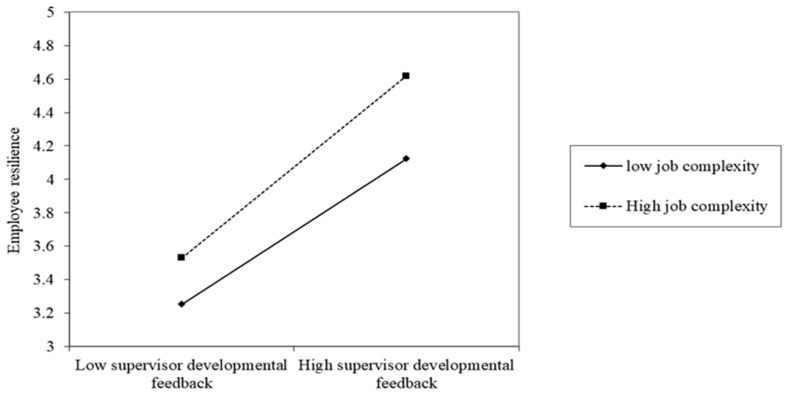
Moderating effect.

**Table 1 ijerph-20-03211-t001:** Basic information of the sample.

Index	Component	Sample	Weight (%)
Gender	Male	148	50.00
	Female	148	50.00
Education	Junior college degree or below	29	9.80
	Bachelor’s degree	175	59.12
	Master’s degree	81	27.36
	Doctoral degree	11	3.72
Age	25 years old and below	68	22.97
	26 to 30 years old	104	35.14
	31 to 35 years old	84	28.37
	36 to 40 years old	25	8.45
	41 years old and above	15	5.07
Tenure	1 to 5 years	166	56.08
	6–10 years	86	29.05
	11–15 years	24	8.11
	16–20 years	9	3.04
	20 years and above	11	3.72

**Table 2 ijerph-20-03211-t002:** Results of confirmatory factor analysis.

	χ^2^	df	χ^2^/df	CFI	TLI	RMSEA	SRMR
Four-factor model (SDF, ER, JC, JS)	102.98	48	2.15	0.97	0.96	0.06	0.05
Three-factor model (SDF, ER + JC, JS)	504.27	51	9.89	0.76	0.69	0.17	0.15
Two-factor model (SDF + ER + JC, JS)	704.71	53	13.30	0.66	0.57	0.20	0.16
Single-factor model (SDF + ER + JC + JS)	1024.55	54	18.97	0.49	0.37	0.25	0.16

Note: SDF, ER, JC, and JS indicate supervisor developmental feedback, employee resilience, job complexity, and job satisfaction, respectively.

**Table 3 ijerph-20-03211-t003:** Descriptive statistics, correlations of variables.

	Mean	SD	1	2	3	4	5	6	7
1. Gender ^a^	0.51	0.51							
2. Age	29.97	6.45	0.00						
3. Education ^b^	2.25	0.68	−0.09	0.01					
4. Tenure	6.35	5.90	0.01	0.92 **	−0.23 **				
5. Supervisor developmental feedback	3.48	0.84	−0.05	0.12 *	0.10	0.13 *			
6. Employee resilience	3.82	0.67	−0.10	0.15 **	0.15 *	0.14 *	0.38 **		
7. Job complexity	3.66	0.81	−0.06	−0.03	−0.12 *	−0.01	0.44 *	0.12 *	
8. Job satisfaction	3.91	0.84	−0.06	0.07	0.14 *	0.04	0.38 **	0.58 **	0.11

Note: n = 296. a. Gender: 0 = male; 1 = female. b. Education: 1 = junior college degree or below; 2 = bachelor’s degree, 3 = master’s degree or higher, 4 = doctoral degree. * *p* < 0.05, ** *p* < 0.01.

**Table 4 ijerph-20-03211-t004:** Results of hierarchical regression analysis.

	Employee Resilience			Job Satisfaction		
	Model 1	Model 2	Model 3	Model 4	Model 5	Model 6
Control variables						
Gender ^a^	−0.11	−0.10	−0.08 **	−0.08	−0.06	0.01
Education ^b^	0.23 **	−0.15 *	0.19	0.19	0.09	−0.01
Age	−0.02	−0.01	−0.02	−0.01 *	0.01	0.02
Tenure	0.04 *	0.02	0.03	0.01	−0.01	−0.03
Independent variable						
Supervisor developmental feedback		0.27 ***	0.29 ***		0.37 ***	0.19 ***
Moderator						
Job complexity			−0.04			
Interaction						
Supervisor developmental feedback*Job complexity			0.08 *			
Mediator						
Employee resilience						0.64 ***
R^2^	0.07	0.18	0.19	0.03	0.15	0.37
adjusted-R^2^	0.05	0.16	0.18	0.01	0.14	0.35
ΔF-statistic	5.01 ***	38.81 ***	5.41 *	1.89	43.82 ***	96.51 ***
ΔR^2^	0.07 ***	0.11 ***	0.02 *	0.03	0.13 ***	0.21 ***

Note: n = 296. a. Gender: 0 = male; 1 = female. b. Education: 1 = iunior college degree or below; 2 = bachelor’s degree, 3 = master’s degree or higher, 4 = doctoral degree. * *p* < 0.05, ** *p* < 0.01, *** *p* < 0.001.

## Data Availability

The raw data generated for this study will be made available by the corresponding author.

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
