# Peer review of "Linking Superior Developmental Feedback with Employee Job Satisfaction? A Conservation of Resources Perspective"

_ijerph, 2023, doi:10.3390/ijerph20043211_

Round 1

Reviewer 1 Report

Overall, this paper is interesting. The statistical approach is appropriate and well described.  A few comments:

Line 14: SDF not defined in abstract

The COVID-19 connection is weak at best.  You state: the primary objective of our research is to explain the association between SDF and J.S. from conservation of resources (COR) theory.  This really has nothing to do with COVID.  People had to have jobs before covid and will certainly after.  In fact, I think removing this part actually makes the paper have more longevity.  I understand you are trying to make the case for a research poor environment due to covid but that case does not need to be made.

Improve figure 1 with labels, specifically: show what is hypothesized to be a mediator and a moderator, not just in the text

Can you explain why you excluded questionnaires that were not completed fully?  What if only one response was missing?

Table 1 is exceedingly challenging to follow, the way it reads (horizontally) is that your male participants are all jr college or below and under 25 and 1 to 5 years old.  This table needs to be horizontal, not vertical.  

Conclusion/discussion: take out parts about covid

Author Response

Thank you for offering us the opportunity to further revise and resubmit our manuscript ijerph-2210829. R1, which is titled “Linking Superior Developmental Feedback with Employee Job Satisfaction? A Conservation of Resources Perspective”.

We would like to thank the review team for your constructive and helpful comments and suggestions. We have made a further attempt to address each of these comments and believe that the manuscript has been substantially improved as a result. We provide point-by-point responses to each specific comment and suggestion given by the review team below.

[Point 1] Overall, this paper is interesting. The statistical approach is appropriate and well described.  A few comments: Line 14: SDF not defined in abstract.

Response: Thank you for pointing this. Following your recommendation, SDF is defined in abstract (line 8).The revised text in the manuscript reads as follows:

 “Abstract: Previous studies have shown that superior superior developmental feedback (SDF) has a mixed impact on employees’ long-term development, but its effect on job satisfaction (JS) has been generally ignored.”

[Point 2] The COVID-19 connection is weak at best.  You state: the primary objective of our research is to explain the association between SDF and J.S. from conservation of resources (COR) theory.  This really has nothing to do with COVID.  People had to have jobs before covid and will certainly after.  In fact, I think removing this part actually makes the paper have more longevity.  I understand you are trying to make the case for a research poor environment due to covid but that case does not need to be made.

Response: Thank you very much. As your suggestion, we removing COVID-19 in the absract. Following is the relevant excerpt from the updated manuscript (please also see line 20-28 in the revised manuscript):

The association between superiors‘ behavior and employees’ role attitude and em-ployee’s psychology has been demonstrated in numerous studies over the past 10 years  [1, 2, 3, 4, 5] (Gilboa, Shirom, Fried, & Cooper, 2008; Jackson & Schuler, 1985; O'Driscoll & Beehr, 1994; Tubre & Collins, 2000). Recently, increasingly researchers have extended superiors‘ behavior, research by examining the employee’s psychology like JS as the outcome [6, 7]. As a crucial component of an employee’s psychology, JS refers to em-ployees’ perceptions of job pleasure as well as their emotional assessments of those perceptions [8, 9]. Communication between superiors and employees is the key to in-fluencing employee JS [2].

[Point 3] Improve figure 1 with labels, specifically: show what is hypothesized to be a mediator and a moderator, not just in the text

Response: Thank you for pointing this. Following your recommendation, h1-h4 is shown in figure 1.

[Point 4] Can you explain why you excluded questionnaires that were not completed fully?  What if only one response was missing?

Response: Thank you very much. We excluded questionnaires that were not matched in two time. To avoid common method bias, we adopted a multi-period data collection method where participants were asked to answer the questionnaires. The total number of questionnaires distributed at Time 1 was 400, and 361 questionnaires were collected. At Time 2, questionnaires were distributed to subjects who completed the questionnaires at Time 1, and 307 questionnaires were recovered. The researchers matched the two questionnaires and finally synthesized the complete questionnaires.

[Point 5] Table 1 is exceedingly challenging to follow, the way it reads (horizontally) is that your male participants are all jr college or below and under 25 and 1 to 5 years old.  This table needs to be horizontal, not vertical. 

Response: Thank you very much. We redraw Table 1 to be horizontal. Please o see line 261 in the revised manuscript

[Point 6] Conclusion/discussion: take out parts about covid

Response: Thanks so much for pointing out this issue. we take out parts about covid in these parts.

Finally, we are grateful for the time and effort you have committed to reviewing our paper and providing us additional constructive feedback. We genuinely believe that our manuscript has improved as a result of the review process. Thank you so much!

Reviewer 2 Report

I have two comments on the article:

- it should be explained why not all questions use a uniform scale (e.g. a five-point Likert scale).

- in the section on reducing staff turnover, attention should be paid to its economic reasons. Appropriate communication and creating development opportunities is of course important. On the other hand, employees often change jobs to raise their income level, which is why the opportunities created on the labor market by other employers are important.

Author Response

Thank you for offering us the opportunity to further revise and resubmit our manuscript ijerph-2210829. R1, which is titled “Linking Superior Developmental Feedback with Employee Job Satisfaction? A Conservation of Resources Perspective”.

We would like to thank the review team for your constructive and helpful comments and suggestions. We have made a further attempt to address each of these comments and believe that the manuscript has been substantially improved as a result. We provide point-by-point responses to each specific comment and suggestion given by the review team below.

[Point 1] it should be explained why not all questions use a uniform scale (e.g. a five-point Likert scale).

Response: Thank you for pointing this. In our reserch, independent variable (SDF), the dependent variable (JS) , the mediating variable (ER)  and  the moderating role of variable (JC) were measured using five-point Likert scales, ranging from 1 (strongly disagree) to 5 (strongly agree). For control variables, Like previous studies (Li et at., 2022; Xia et at., 2017), we code education and age as categorical variables instead of a dummy.

“Li, Z.Q., Ma, C., Zhang, X., Guo, Q.M. (2022).Full of energy – The relationship between supervisor developmental feedback and task performance: a conservation of resources perspectives. Personnel Review, DOI10.1108/PR-03-2021-0138

Xia,Y., Zhang, L., & Li, M.. (2017). Abusive leadership and helping behavior: capability or mood, which matters?. Current Psychology. DOI 10.1007/s12144-017-9583-y

[Point 2] - in the section on reducing staff turnover, attention should be paid to its economic reasons. Appropriate communication and creating development opportunities is of course important. On the other hand, employees often change jobs to raise their income level, which is why the opportunities created on the labor market by other employers are important.

Response: Thank you very much for your valuable comments. In order to avoid ambiguity, we have deleted the description of staff turnover, focusing on why and when superior developmental feedback can influence employee job satisfaction based on cognitive dissonance theory onservation of resources (COR) theory.

Finally, we are grateful for the time and effort you have committed to reviewing our paper and providing us additional constructive feedback. We genuinely believe that our manuscript has improved as a result of the review process. Thank you so much!

Reviewer 3 Report

Congratulations for the article. It really is a very important theme and seems to need more in-depth studies like this one. The satisfaction of employees and the feedback given to them help to improve productivity levels within companies, and this was proven by the article. As the study was applied in the Chinese context, it is important that it be applied in other regions and countries as well, to compare the results and create actions to disseminate the knowledge to the world. So, my question is to know how the questionnaire was structured: from the literature or was it adapted from a previous study? Was this questionnaire evaluated before being applied to employees? It would be important to have elements about this in the methodology, so that this study can be replicated.

Author Response

Response:

Thank you for offering us the opportunity to further revise and resubmit our manuscript ijerph-2210829. R1, which is titled “Linking Superior Developmental Feedback with Employee Job Satisfaction? A Conservation of Resources Perspective”.

We would like to thanks for your constructive and helpful comments and suggestions. We have made a further attempt to address each of these comments and believe that the manuscript has been substantially improved as a result.

Superior developmental feedback (Zhou, 2003), employee resilience (Smith et al., 2008), job satisfaction(Shaw & Gupta, 2008) and job complexity(Liu et al., 2007) used in this study were derived from previous research published in reputable journals, guaranteeing their accuracy. All scale were measured using five-point Likert scales, ranging from 1 (strongly disagree) to 5 (strongly agree). s In order to ensure the accuracy of the questionnaire, we used translation - back translation procedure (Brislin, 1980).Two bilingual doctoral students were chosen to translate the English questionnaire into Chinese, and later, one lecturer and two doctorate students were selected to translate the Chinese questionnaire back into English. After completing these steps, 10 employees from different companies were asked to participate in a pre-survey to ensure the accuracy of the questionnaire expression.

Brislin, R. W. (1980). Translation and content analysis of oral and written materials. In H. C. Triandis & J. W. Berry (Eds.), Handbook of cross-cultural psychology: Methodology: 389–444. Boston: Allyn and Bacon.

Liu, C., Spector, P. E., Shi, L. Cross‐national job stress: a quantitative and qualitative study. J. Organ. Behav. 2007, 28(2), 209-239. https://doi.org/10.1002/job.435.

Shaw, J.D., Gupta, N. Job complexity, performance, and wellbeing: when does supplies-values fit matter? Pers. Psychol. 2004, 57(4), 847-879. https://doi.org/10.1111/j.1744-6570.2004.00008.x.

Smith B.W., Dalen J., Wiggins K., Tooley, E., Christopher, P., Bernard, J. The Brief Resilience Scale: Assessing the Abil-ity to Bounce Back. Int. J. Behav. Med. 2008, 15(3), 194-200. https://doi.org/10.1080/10705500802222972.

Zhou, J. When the presence of creative coworkers is related to creativity: Role of supervisor close monitoring, devel-opmental feedback, and creative personality. J. Appl. Psychol. 2003, 88, 413-422. https://doi.org/10.1037/0021-9010.88.3.413.

Finally, we are grateful for the time and effort you have committed to reviewing our paper and providing us additional constructive feedback. We genuinely believe that our manuscript has improved as a result of the review process. Thank you so much!